# 2,4-Thiazolidinedione in Well-Fed Lactating Dairy Goats: II. Response to Intra-Mammary Infection

**DOI:** 10.3390/vetsci6020052

**Published:** 2019-06-05

**Authors:** Fernanda Rosa, Misagh Moridi, Johan S. Osorio, Jayant Lohakare, Erminio Trevisi, Shelby Filley, Charles Estill, Massimo Bionaz

**Affiliations:** 1Department of Animal and Rangeland Sciences, Oregon State University, Corvallis, OR 97331, USA; Fernanda.Rosa@sdstate.edu (F.R.); Johan.Osorio@sdstate.edu (J.S.O.); shelby.filley@oregonstate.edu (S.F.); Charles.Estill@oregonstate.edu (C.E.); 2Department of Animal Science, University of Guilan, Kilometer 5 of Rasht-Qazvin Highway, Rasht 4199613776, Iran; moridimisagh64@gmail.com; 3Department of Animal Biotechnology, Kangwon National University, KNU Ave 1, Chuncheon 200-701, Korea; lohakarej@uapb.edu; 4Department of Animal Sciences, Food and Nutrition (DIANA), Università Cattolica del Sacro Cuore, Via Emilia Parmense, 84, 29122 Piacenza PC, Italy; erminio.trevisi@unicatt.it

**Keywords:** 2,4-thiazolidinedione, goat, mastitis, macrophages, mammary epithelial cells

## Abstract

In a prior experiment, treatment of goats with the putative PPARγ agonist 2,4-thiazolidinedione (2,4-TZD) ameliorated the response to intramammary infection without evidence of PPARγ activation. The lack of PPARγ activation was possibly due to deficiency of vitamin A and/or a poor body condition of the animals. Therefore, the present study hypothesized that activation of PPARγ by 2,4-TZD in goats supplemented with adequate amounts of vitamin A can improve the response to sub-clinical mastitis. Lactating goats receiving a diet that met National Research Council requirements, including vitamin A, were injected with 8 mg/kg BW of 2,4-TZD (*n* = 6) or saline (*n* = 6; control (CTR)) daily. Two weeks into treatment, all goats received *Streptococcus uberis* (IMI) in the right mammary gland. Blood biomarkers of metabolism, inflammation, and oxidative status plus leukocytes phagocytosis were measured. Mammary epithelial cells (MEC) and macrophages were isolated from milk and liver tissue collected for gene expression analysis. Milk fat was maintained by treatment with 2,4-TZD, but decreased in CTR, after IMI. Haptoglobin was increased after IMI only in 2,4-TZD without any effect on negative acute phase proteins, indicating an improved liver function. 2,4-TZD vs. CTR had a greater amount of globulin. The expression of inflammation-related genes was increased by IMI in both macrophages and MEC. Except for decreasing expression of *SCD1* in MEC, 2,4-TZD did not affect the expression of measured genes. Results confirmed the successful induction of sub-clinical mastitis but did not confirm the positive effect of 2,4-TZD on the response to IMI in well-fed goats.

## 1. Introduction

Mastitis is a multifactorial disease that affects dairy animals resulting from an inflammation of the mammary gland, mostly caused by pathogens [1]. It is the most costly disease in the dairy industry with negative consequences to animal well-being and milk quantity and quality [2]. Mastitis can be treated using antibiotics, but the possibility of antibiotic resistance makes this practice dangerous, and consumers are requesting that producers avoid or minimize the use of antibiotics [3]. Therefore, alternatives to antibiotics are in high demand, and approaches to mastitis prevention are a priority for the dairy industry.

Practices to aid in preventing mastitis include the maintenance of animals in a clean environment and pre- and post-dip procedures performed during milking. Nutrition plays a key role in mastitis prevention. Among nutrients, an adequate level of lipophilic vitamins aids in decreasing mastitis prevalence [4]. Vitamin A and β-carotene are important in mastitis prevention due to the stimulatory effects on immune cells [5]. A significant inverse association between the risk of clinical mastitis and serum retinol (vitamin A) were observed in Holstein cows [6]. In addition, β-carotene can acts as an antioxidant, reducing the damage by superoxide produced during phagocytosis [7].

Nutrition can also help prevent mastitis by a nutrient-gene interaction (a.k.a. nutrigenomics), by which dietary compounds can affect the animal’s metabolism and performance via interaction with transcription factors [8]. Clear nutrigenomic roles have been established for ligand-dependent nuclear receptors, especially the peroxisome proliferator-activated receptors (PPAR). These are known to be activated by synthetic ligands, such as thiazolidinedione and fenofibrate, and natural ligands, such as fatty acids [9].

Among PPAR, the isotype PPARγ has been well-studied in monogastrics playing important anti-inflammatory roles [10]. In our prior experiment [11], the use of the putative PPARγ agonist 2,4-thiazolidinedione (2,4-TZD) in dairy goats with induced sub-clinical intramammary infection improved overall liver function and increased the level of myeloperoxidase in blood, i.e., the killing capacity of neutrophils. In that experiment, goats received hay without adequate supplementation of vitamins. Work from our lab demonstrated that 9-*cis*-retinoic acid (a metabolite of vitamin A) is essential for the *in vitro* activation of PPAR by 2,4-TZD in bovine and goat mammary cells [8].

Based on above, we hypothesized that treatment of lactating dairy goats with 2,4-TZD, while providing sufficient vitamin A supplementation, would improve the response to a subclinical mammary infection via the activation of PPARγ. The objectives of this study were to test whether 2,4-TZD-induced PPARγ activation in dairy goats receiving an adequate amount of vitamin A 1) improves the metabolic and inflammatory response to induced subclinical mammary infection and 2) affects the expression of target genes in liver, mammary epithelial cells (MEC), and macrophages.

## 2. Materials and Methods

### 2.1. Experimental Design and Animal Management

The present experiment is part of a larger project where the same animals were used to accomplish two main objectives. The first objective was to assess the effect of 2,4-TZD on milk fat synthesis [12] and the second objective was to assess the effect of 2,4-TZD on the response of goats to intramammary infection (present manuscript). See the companion manuscript [12] for overall experimental design and animal management. Briefly, the experimental design involved injecting the treated goats daily with 8 mg of 2,4-TZD/kg of BW dissolved in 10 mL of sterile saline (TZD; *n* = 6) and injecting the same amount of saline only to control goats (CTR; *n* = 6) until the end of the trial. For the first 10 days of the experiment, each goat nursed kids.

To induce subclinical mastitis, after two weeks of daily i.v. injections of 2,4-TZD or saline, all goats received an intramammary infection (IMI) with an infusion of 1.7 × 10^8^ cfu of *Streptococcus uberis* (*Strep. uberis*) in 10 mL sterile physiological saline (002479, Henry Schein, Melville, NY, USA) in the right half of the mammary gland. The left half of the mammary gland was left untouched and used as control. The choice of not infusing the left gland with saline was due to its previously observed induction of slight inflammation when infused in the mammary gland [11]. The pre-dosed aliquots of *Strep. uberis* (strain 0140J, see [11]) in 1.5 mL sterile vials were provided by the laboratory of Peggy Dearing, College of Veterinary Medicine, Oregon State University. Prior to IMI, the teat ends were carefully cleaned with individual moistened towels and disinfected with swabs containing 70% ethanol. The infusion was performed with the aid of a disposable sterile TOMCAT urinary open 3.5fr catheter. After infusion, the inoculate was thoroughly massaged upward into the gland cistern for 30 s.

After IMI, there were 4 groups for all parameters measured in milk plus MEC and macrophages:-CTR-control, untouched left mammary gland from goats receiving an intrajugular injection of saline;-CTR-IMI, right mammary gland infused with *Strep uberis* from goats receiving an intrajugular injection of saline;-TZD-control, untouched left mammary gland from goats receiving an intrajugular injection of 2,4-TZD;-TZD-IMI, right mammary gland infused with *Strep uberis* from goats receiving an intrajugular injection of 2,4-TZD.

For blood parameters and gene expression in the liver, only two groups were available: TZD and CTR.

### 2.2. Measurements, Sample Collection, and Blood Metabolites

Materials and methods regarding dry matter intake, milk analyses, and isolation of MEC from the milk are detailed in the companion manuscript [12].

Milk was collected aseptically from both halves of the mammary gland for bacteriological analysis 3 days prior to IMI. Before sampling, teats were cleaned using iodine solution and disposable paper towels. The orifice of the teat was disinfected with swabs containing 70% ethanol. Approximately 1 mL of milk was collected in sterile 1.5 mL tubes. The samples were immediately placed on ice and shipped overnight to Ag Health Laboratories, Inc. (Sunnyside, WA, USA) for a milk bacterial culture on Bovine blood agar plates. All the samples were negative for any pathogen. To ascertain the status of the mammary gland a California Mastitis Test to estimate milk somatic cell count (SCC) was performed at the last milking prior to IMI.

Goats were milked using a portable machine with two claws. To avoid cross-contamination, before IMI each goat was milked using one claw; after IMI both claws were used to milk one goat with one claw exclusively used to milk the right gland (i.e., the infected half) and the other claw was used to milk the healthy gland (Appendix A) of each goat. Milk yield from both halves at each milking was recorded, and milk samples were collected just before IMI, at 12 h post-IMI, and then at 1, 2, 5, 6, and 7 days post-IMI as described in the companion manuscript [12].

Rectal temperature (RT) was checked using a digital rectal thermometer prior to IMI, 12 h post-IMI and subsequently twice a day (A.M. and P.M. milking) until the end of the experiment.

### 2.3. Blood Metabolites and Inflammatory Markers

Collection, handling, and analysis of blood samples were as described in the companion manuscript [12]. For the present objective, samples were also collected at 12 h post-IMI and the following parameters were also measured besides glucose, non-esterified fatty acids (NEFA), triglycerides (TG), β-hydroxybutyrate (BHBA), cholesterol, creatinine, and urea [12]: inflammatory-related parameters albumin, haptoglobin, ceruloplasmin, paraoxonase, myeloperoxidase, and total bilirubin; the oxidative-related parameters total reactive oxygen metabolites (ROM), ferric reducing antioxidant power (FRAP) and Zn; plus aspartate aminotransferase, gamma-glutamyl transferase, and total protein. In addition, globulin was obtained by subtracting albumin from total protein [13]. These analyses were performed following the procedures described previously [13,14] using a clinical auto-analyzer (ILAB 650, Instrumentation Laboratory, USA Spa, Werfen Co., Milan, Italy). The intra- and inter-assay variation for the parameters measured are available in Appendix A.

### 2.4. Phagocytosis and Percent Leukocytes

The proportion of granulocytes and monocytes and their phagocytosis were performed prior to IMI and, to have a similar time point as MEC and macrophages isolation (see below), 6 d post-IMI as previously described [11].

### 2.5. Macrophages Isolation from Milk

Macrophages were isolated from 250 mL of milk from goats at 5 d post-IMI. Milk was collected in 250 mL sterile tubes (430776, VWR, Radnor, PA, USA) and immediately placed on ice (4 °C) until isolation (~1 h). The isolation was performed using magnetic sorting with a monoclonal antibody against CD14 (CAM36A, CT-BOV2027, Monoclonal antibody center, Pullman, WA, USA) [15]. We obtained 5.5 × 10^6^ ± 4.9 × 10^6^ macrophages (22,260 macrophages/mL of milk) as measured using MoxiZ (Orflo, Ketchum, ID, USA). The evaluation of the enrichment of macrophages after isolation was assessed via RT-qPCR by measuring the expression of the cluster of differentiation 14 (*CD14*) in positive and negative isolated cells.

### 2.6. Euthanasia and Tissue Collection

At the end of the trial, nine goats (6 in TZD group and 3 in CTR group) were euthanized by rapid intravenous injection of a barbiturate (Somnasol euthanasia solution, Henry Schein) to harvest liver for gene expression analysis. The liver was harvested using a sterile scalpel and forceps. The tissue was immediately cleaned with RNase decontamination solution (RNaseZap^®^, AM9780, Thermo Fisher Scientific, Waltham, MA, USA), quickly blotted with sterile gauze to remove residual blood, and snap-frozen in liquid nitrogen. Samples were then stored at −80 °C until RNA extraction.

### 2.7. RT-qPCR

RNA isolation and RT-qPCR protocol are described in the companion manuscript [12]. The RNA integrity number (RIN) values had a mean ± SD of 7.4 ± 0.7 for liver, 7.0 ± 1.2 for macrophages, and 6.9 ± 1.2 for MEC samples. The target genes selected to be evaluated in liver RNA samples were PPARγ (*PPARG*), pyruvate dehydrogenase kinase 4 (*PDK4,* a PPARβ/δ target gene), carnitine palmitoyl transferase 1A (*CPT1A*, a PPARα target gene), apolipoprotein E (*APOE,* a PPARγ target gene), the acute phase protein haptoglobin (*HP*), and glyceraldehyde 3-phosphate dehydrogenase (*GAPDH*). For MEC we measured transcription of inflammatory-related genes interleukin 8 (*IL8*), chemokine (C-C Motif) ligand 2 (*CCL2*), nuclear factor of kappa light polypeptide gene enhancer in B-cells 1 (*NFKB1*), and tumor necrosis factor alpha (*TNFA*). In addition, the transcript abundance of milk fat synthesis-related genes stearoyl-CoA desaturase (delta-9-desaturase) (*SCD1*) and cluster of differentiation 36 (*CD36*) were measured in MEC. For macrophages the transcripts evaluated were related to general inflammatory response or classical macrophages activation (*IL8, CCL2, NFKB1*, *TNFA,* nitric oxide synthase 2 (*NOS2*)), alternative macrophages activation (arginase 1 (*ARG1*), Mannose Receptor C-Type 1 (*MRC1*), and interleukin 4 (*IL4*)), and transforming growth factor beta 1 (*TGFB1*) and interleukin 10 (*IL10*) as markers of macrophages inactivation [16]. In addition, small ubiquitin-like modifier 1 (*SUMO1*), *CD14*, *PPARG, CD36,* and *GAPDH* were measured in macrophages. Primer-pairs were designed as previously described [11]. Amplicons not previously validated were assessed as previously described [12]. The details of primer pairs and amplicon sequences are available in Appendix A.

The genes ribosomal protein 9 (*RPS9*), ubiquitously-expressed transcript (*UXT*) and tyrosine 3-monooxygenase/tryptophan 5-monooxygenase activation protein, zeta polypeptide (*YWHAZ*) were used as internal control genes for liver and macrophages. The reliability of the normalization using the above three reference genes was determined by the V-value [17] and was 0.04 for liver and 0.13 for macrophages. *RPS9*, *YWHAZ,* and *GAPDH* were used to normalize RT-qPCR data for MEC (V-value = 0.19).

### 2.8. Statistical Analysis

Blood parameters from the entire experiment were analyzed as described in the companion manuscript [12]. Prior to statistical analysis, all data except percent phagocytosis and gene expression were arithmetically transformed to have the same mean at time 0 relative to IMI (i.e., baseline) for TZD and CTR. This was achieved by subtracting from each data point the difference between TZD and CTR at time 0 relative to IMI [12].

For milk data, the correction was done relative to the left mammary gland of the CTR group using the same approach as above. Detection of outliers and check for normal distribution were as performed in the companion manuscript [12]. Data were analyzed with the PROC GLIMMIX procedure of SAS 9.4 (SAS Institute, Inc., Cary, NC, USA). Fixed effects in the model were treatment (i.e., TZD or CTR), time relative to 2,4-TZD or IMI, and all interactions. For milk yield and milk composition, SCC and MEC and macrophage gene expression, IMI was also included in the main effect (including interactions). The goat was considered a random effect in all analyses. The best covariance structure was selected as described in the companion manuscript [12]. Significance and tendencies were declared at *p* ≤ 0.05 and 0.05 < *p* ≤ 0.10, respectively.

## 3. Results

### 3.1. Body Temperature and Feed Intake

Dry matter intake was not affected by IMI or 2,4-TZD while rectal temperature increased within 12 h post-IMI and remained higher than pre-IMI for the first 5 days post-IMI (Figure 1). Goats treated with 2,4-TZD had an overall greater temperature compared to CTR goats, especially during the first 5 days post-IMI.

### 3.2. Milk yield, SCC and Milk Composition

Milk yield was affected by time and tended (*p* = 0.07) to be lower in the mammary gland receiving IMI (Figure 2). Milk yield was higher in TZD-control compared with TZD-IMI during the first 3 days post-IMI, i.e., a significant TZD × IMI interaction. SCC was increased and remained ≥10-fold higher in IMI than non-IMI mammary glands until the end of the trial (Figure 2). SCC during the whole experiment had a tendency (*p* = 0.06) to be affected by the interaction TZD × Time, due to a 3-fold lower SCC in CTR vs. TZD in the first week after starting 2,4-TZD injection (Appendix A).

Milk fat percentage was affected by 2,4-TZD in a time-dependent manner (TZD × Time) with 8% larger fat percentage in TZD compared to CTR (Figure 3). Lactose and protein percentages were affected by the interaction Time × IMI, where IMI decreased lactose and increased protein during the first 2 days post-IMI (Figure 3). Lactose yield tended to decrease (*p* = 0.08), and fat yield was decreased by IMI.

### 3.3. Blood Biomarkers

The pattern of glucose, NEFA, TG, BHBA, cholesterol, creatinine, and urea concentration in plasma was not affected by 2,4-TZD treatment after IMI, but several parameters were affected by Time (i.e., IMI) (Table 1). IMI decreased the concentrations of TG and increased the concentration of glucose, reaching a peak at 12 d post-IMI.

Among positive acute phase proteins, haptoglobin was overall lower (Figure 4) in TZD vs. CTR during the entire study. However, in the early response to IMI, haptoglobin concentration tended (*p* = 0.06) to increase 1.4-fold more in TZD vs. CTR. An interaction TZD × Time was detected for albumin, globulin, bilirubin, and ROM (Figure 4 and Figure 5) during the entire study.

TZD vs. CTR had higher globulin before IMI and tended (*p* = 0.06) to have larger ROM after IMI. The level of bilirubin was lower early after IMI and was larger 11 days post-IMI in TZD vs. CTR. The concentration of Zn tended (*p* = 0.10) to be lower in TZD vs. CTR (Figure 5). No significant effects, except a TZD × Time for total protein, were detected for other inflammatory, metabolic, and antioxidative-related parameters (Appendix A).

### 3.4. Differential Leukocyte Count and Phagocytosis

The percentage of granulocytes was not affected by IMI, while the percentage of monocytes tended (*p* = 0.06) to decrease after IMI (Figure 6). Monocytes in TZD had greater phagocytosis compared to CTR (Figure 6).

### 3.5. Gene Expression in Liver and MEC

None of the measured genes in the liver, including the known targets of the three PPAR isotypes, were transcriptionally affected by 2,4-TZD (Appendix A). In MEC (Figure 7), transcription of all measured pro-inflammatory related genes was ≥2-fold up-regulated by IMI, and only the expression of *CCL2* had a tendency (*p* = 0.10) to be lower in TZD vs. CTR. The expression of *SCD1* was decreased >3-fold by IMI and was lower in TZD vs. CTR. The expression of *CD36* tended (*p* = 0.08) to be decreased by IMI but due to a >2-fold decrease only in CTR after IMI (IMI × TZD; *p* = 0.09).

### 3.6. Gene Expression in Macrophages

In macrophages isolated from the milk, the expression of *CD14* increased approximately 3.5-fold, and the enrichment of CD14-expressing cells did not affect the expression of cytokeratin 8 (Appendix A).

Compared to MEC, the CD14-positive cells (i.e., macrophages) had a higher mRNA abundance of the measured pro-inflammatory-related genes *NFKB1*, *TNFA*, *CCL2*, and *IL8* (Appendix A).

2,4-TZD did not affect the expression of any measured genes in macrophages (Figure 8). Compared to control, IMI increased the expression of the measured genes related to the classical macrophage activation (i.e., *NFKB1*, *TNFA*, *IL8,* and *CCL2*) as well as the expression of *TGFB1* and the expression of genes related to macrophage deactivation, *CD14*, *GAPDH*, and *NOS2* (Figure 8). The expression of markers for the alternative activation of macrophages was either not affected by IMI (*MRC1*; Figure 8) or was below the limit of detection in most of the samples (*ARG1* and *IL4*).

## 4. Discussion

### 4.1. Response to IMI in Comparison with the Prior Study

The present experiment was a repeat of a previously published experiment [11]. However, the two experiments present important differences. Compared to the prior experiment, the present experiment had: (a) goats in an earlier stage of lactation (early–mid vs. late lactation); (b) well-fed vs. poorly-fed goats; (c) initial body condition score of 2.6 vs. 1.6 (1–5 scale); d) IMI in one mammary gland vs. both glands; (e) the gland kept as control did not receive any infusion vs. infusion of sterile saline; (f) 2,4-TZD was injected for 2 weeks vs. 1 week prior IMI; (g) 2,4-TZD was injected daily at 10:00 a.m. vs. 12:00 a.m.; (h) goats were milked twice a day vs. one milking per day (3.6 vs. 1 kg/d); and (i) goats were individually fed vs. group fed. Despite the above differences, it is important to compare the two studies.

Similar to what was observed in the prior experiment [11], in the present experiment, IMI induced a consistent increase in SCC that remained higher than the control gland until the end of the trial. The SCC was high with no clinical sign of mastitis, milk yield changed, although for a shorter period of time (first 3 days post-IMI), and in a less pronounced amount in the present compared to the prior experiment, where the reduction of milk yield was detected until the end of the trial in the goats receiving *Strep uberis*. This can be partly due to the infection being induced in only one of the two halves of the mammary gland in the present experiment compared to both mammary halves in the prior experiment. Other indicators of a temporary inflammation in our experiment were the spike in glycemia, the decrease of Zn [18], and the increase in haptoglobin [13,19]. Furthermore, we detected an increase in expression of most of the measured pro-inflammatory related genes in MEC and macrophages, indicating a robust inflammatory response. However, the response was smaller compared to the prior experiment with little-to-no effect on leukocytes, especially the percentage of PMN, which was decreased in the prior experiment [11]. Despite the differences compared to the prior experiment, all of the above data are supportive of successful induction of subclinical mammary infection in the present experiment.

### 4.2. Treatment with 2,4-TZD Allows Maintaining Milk Fat Synthesis after IMI but Not through SCD1

More evident in the present experiment compared to the prior experiment [11] was the maintenance of milk fat percent after IMI in TZD vs. CTR. This observation appears to partly support the proposed role of PPARγ in controlling milk fat synthesis [8,9]; however, *SCD1*, a putative PPARγ target gene [9], was lower in TZD vs. CTR. The above data indicate that either this gene is not a PPARγ target gene or that PPARγ was not activated by 2,4-TZD, as supported by the lack of change in expression of the putative PPAR target gene *CD36* in MEC and the findings reported in the companion manuscript [12]. The down-regulation of *SCD1* despite the maintenance of milk fat production further support the indication from the companion manuscript [12] of a not essential role of SCD1 for milk fat production in goats.

In goats receiving a saline injection, milk fat was decreased in both mammary glands after IMI, the one with infection and the healthy gland. This is consistent with previously observed detrimental effect on milk synthesis of mastitis in one gland to the other adjacent mammary glands [19]. However, milk fat did not decrease in the milk from the control gland in the TZD group. The reason for such “uncoupling” in our experiment is unclear, but it appears that 2,4-TZD prevented any negative consequence of IMI on milk fat synthesis in the healthy gland.

### 4.3. Treatment with 2,4-TZD Modestly Improves the Liver Response to Mammary Infection

The response of haptoglobin to IMI was different between the two experiments. In the prior experiment, the increase in haptoglobin after IMI was more pronounced, and it remained high (i.e., >0.3 g/L) until the end of the trial with a somewhat similar pattern in all animals [11]; in the present experiment the change in haptoglobin after IMI was relatively modest and with a quick spike only in TZD group. The results of the present experiment are somewhat similar to another study where goats received an intramammary infusion of *Staphilococcus aureus* in one of the two halves of the mammary gland [20], although in that experiment haptoglobin returned to baseline after 6 days instead of 3 days post-IMI.

None of the negative acute phase proteins were different between the two groups in the present experiment, while differences were observed in the prior experiment [11]. Bilirubin is cleared by the liver, and an increase in the blood can be ascribed to a decreased liver function, as was already known at the beginning of the 20th century [21]. Thus, the bilirubin level in blood is also an important index for the evaluation of liver function in animals experiencing inflammatory-like conditions [13,14]. The detection of numerically lower bilirubin in TZD vs. CTR despite a higher level of haptoglobin during the first 3 days post-IMI is indicative of the improved ability of the liver in TZD vs. CTR to handle the negative consequence of an induced inflammation [22]. An improved liver condition after TZD was also among the major findings in the prior experiment [11], although the effect was more pronounced compared to the present experiment.

From the above data, it is clear that the effect of 2,4-TZD in the liver in the present experiment was very modest compared to the prior experiment [11]. This is also supported by the lack of any change in measured transcripts in the liver, although the number of liver samples in the CTR group was limited (*n* = 3).

Overall, the modest metabolic and inflammatory responses detected in the present experiment after IMI compared to the prior experiment is indicative of a better, more robust, faster, and more effective response to mastitis in the animals used in the present vs. the prior experiment likely due to better nutrition and body condition.

### 4.4. Treatment with 2,4-Thiazolidinedione Increases Plasma Globulin and Increases Monocyte Phagocytosis

Goats treated with 2,4-TZD had an increased production of globulin compared to CTR. Globulin in the blood is made up of four fractions; however, change in plasma globulin is associated with a change in the level of immunoglobulins [23]. Increase proportion of immunoglobulin is often associated with health disorders in humans [24], and higher globulin level has been associated with poor udder health in cows with high SCC in milk but with no bacterial growth in culture [25]. In our experiment, we did observe a tendency for higher SCC in TZD vs. CTR during the first few days of 2,4-TZD injection, mostly due to a decrease in the CTR; however, the lower level of haptoglobin in TZD vs. CTR is indicative of lack of any sign of inflammation. Therefore, the higher globulin level remains unexplained.

Our data indicated significantly larger phagocytosis in monocytes in goats treated with 2,4-TZD compared to control. Activation of PPARγ in human increases phagocytosis in monocytes [26]. PPARγ activation by curcumin and consequent up-regulation of *CD36* was also associated with higher human monocytes phagocytosis [27]. It is difficult to infer if 2,4-TZD activated PPARγ in monocytes in our study. We did not measure the expression of *CD36* in blood monocytes; however, its expression was not affected by 2,4-TZD in milk macrophages, but, different than the control goats, *CD36* expression tended to be not affected by IMI in MEC. Furthermore, none of the other data indicated the activation of PPARγ by 2,4-TZD. Therefore, it is still possible that 2,4-TZD affected monocytes phagocytosis via activation of PPARγ but very unlikely. Overall, increase phagocytosis by monocytes can be a favorable effect for goats due to the larger persistence and ability to respond to stimuli compared to neutrophils [28].

### 4.5. Treatment with 2,4-TZD has a Moderate Effect on Oxidative Parameters

Despite never reaching a statistical significance in any of the time points collected, the goats treated with 2,4-TZD had numerically greater ROM. The reason for such an increase is unclear. It is plausible that the higher amount of ROM was caused by a greater catabolic state in TZD vs. CTR; this is somewhat supported by the detected greater rectal temperature in TZD vs. CTR. The numerically higher ROM had likely no biological effect considering the lack of any detected difference in the measured antioxidant parameters compared to CTR.

### 4.6. IMI but Not 2,4-TZD Affects the Transcriptome of MEC and Macrophages

As MEC and resident macrophages in the mammary gland are the first line of defense against mastitis [29], as expected, the transcripts measured in MEC and macrophages coding for proteins with main inflammatory/immune response functions were all increased by IMI. These findings are in agreement with other studies that demonstrated up-regulation of genes coding for proteins that stimulate the chemotaxis of neutrophils and monocytes, such as *IL8* and *CCL2*, upon intramammary infection in bovine mammary tissue [30,31]. However, 2,4-TZD did not affect the expression of the inflammation-related genes measured, with the exception of a tendency in TZD vs. CTR for a lower expression of *CCL2,* a major chemoattractant of monocytes, macrophages, and lymphocytes [32].

In rodents, PPARγ plays an important role in preventing macrophage activation [33]. Further, PPARγ activation prevents the production of inflammatory cytokines in human monocytes [34]. Macrophages can be activated through the classical and the alternative activation pathways [16]. TNFα and inducible nitric oxide synthase are markers of the classical activation of macrophages stimulated by interferon gamma, while arginase 1 and dectin 1 are markers of the alternative activation induced by interleukins 4 and 13. The macrophage deactivation is initiated by IL10 and TGFβ, which in turn induce the production by macrophages of IL10 and TNFβ [16]. The use of a PPARγ agonist induced the alternative activation of macrophages but also induced macrophages deactivation in mice [35]. In our study, we did not observe any effect of 2,4-TZD on the above markers.

The activity of PPARγ can also be modulated by SUMOylation via small ubiquitin-related modifier 1 [36]. The SUMOylation generally has a negative effect on the activation of PPARγ by repressing its basal and induced activation [37]; however, the activation of PPARγ by ligands can prevent SUMOylation [37]. The repression of the inflammatory response of macrophages in mice involves, as the first step, a SUMOylation of the PPARγ ligand domain [38]. Therefore, it is possible that the lack of response to 2,4-TZD by macrophages in our experiment was due to an increase in SUMOylation that could have prevented the binding of the 2,4-TZD to the ligand binding domain of the PPARγ. However, we did not detect any effect of 2,4-TZD on the transcription of *SUMO1,* the gene coding for SUMOylation proteins with a major role in the transcriptional activation of PPARγ [39]. Thus, it remains unclear if SUMOylation played a role in preventing PPARγ response to 2,4-TZD in our experiment.

Overall, our data indicated that 2,4-TZD did not affect the inflammatory response of macrophages and MEC.

## 5. Conclusions

Treatment of goats with 2,4-TZD had a modest effect on the immune system with a slight increase in liver function. However, the magnitude of the effects observed was lower compared to goats receiving the same treatments but in less-then-optimal conditions [11]. A lack of response could be due to the balanced ration used in the present experiment. The optimal nutritional status might have already maximized the activation of PPARγ. If this were true, it would suggest that interventions to modulate transcription factors might be more beneficial during a state of nutritional deficiency. However, it is very unlikely that PPARγ cannot be further activated if a potent agonist is used. The lack of effects by 2,4-TZD on the expression of PPAR target genes in the liver, MEC, and macrophages despite optimal supplementation of vitamin A confirms the inability of 2,4-TZD to activate PPARγ in vivo.

The lack of activation of PPARγ by 2,4-TZD despite the goats being in optimal condition and well-fed prevented the testing of our hypothesis that activation of PPARγ can aid in the response to intramammary infection. Despite not activating PPARγ, 2,4-TZD treatment allowed to preserve milk fat percentage after IMI. The mechanism of such an effect remains elusive. A study using true PPARγ agonists in goats should be performed to test the original hypothesis.

## Figures and Tables

**Figure 1 vetsci-06-00052-f001:**
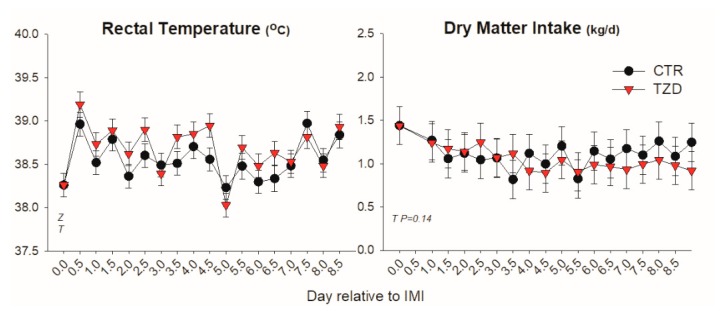
Rectal temperature and dry matter intake (kg/d) in Saanen goats treated with 2,4-thiazolidinedione (TZD) or saline (CTR) after intramammary infusion of *Strep. Uberis*. Letters in the graph denote significant (*p* < 0.05) effects of time (T), mastitis (M), 2,4-thiazolidinedione treatment (Z), and interactions (Z × T). The reported *p*-value is for tendencies.

**Figure 2 vetsci-06-00052-f002:**
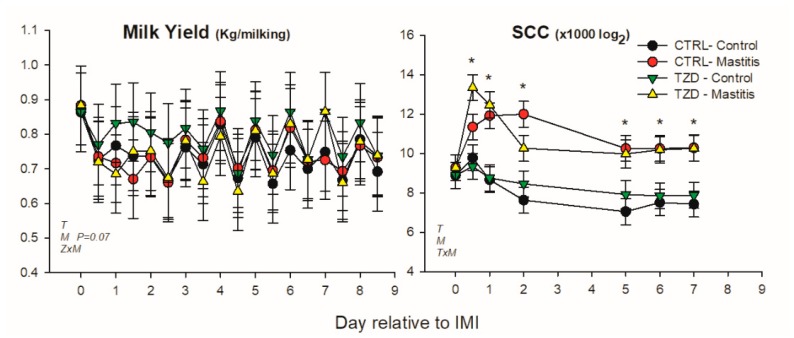
Effect on milk yield and somatic cell count (SCC) of a daily injection of 2,4-thiazolidinedione (TZD) and saline (CTR) in *Saanen* dairy goats plus intramammary infusion in the right half of the mammary gland with *Strep. uberis* (IMI) while the left half of the mammary was left untouched (control). Overall statistical significance (*p* ≤ 0.05) is indicated in the plot as follows: T = time effect; M = IMI effect; Z = effect of 2,4-thiazolidinedione injection. When a significant (*p* ≤ 0.05) interaction was present for T × M, the difference between groups at each time point is denoted by *.

**Figure 3 vetsci-06-00052-f003:**
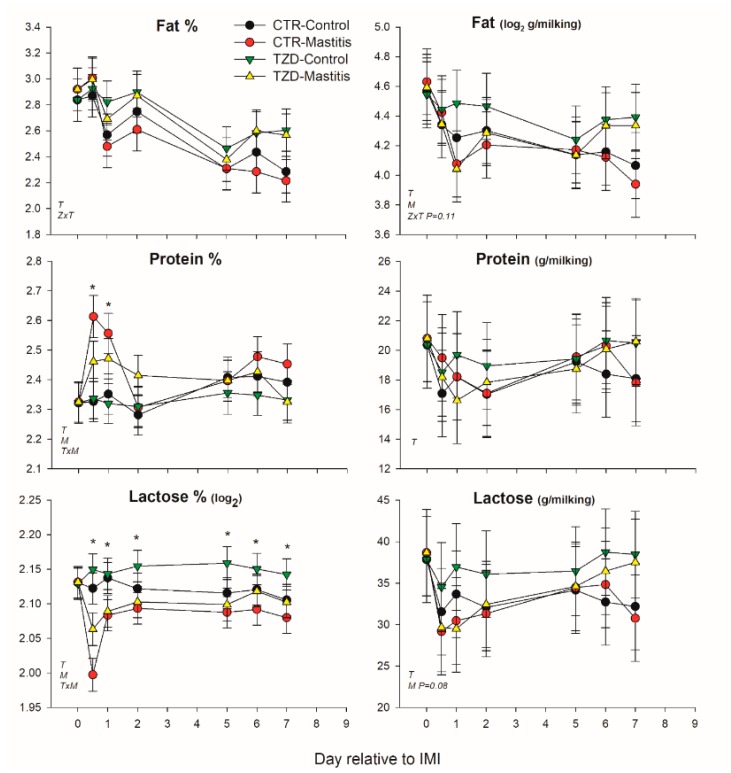
Effect on milk components of a daily injection of 2,4-thiazolidinedione (TZD) and saline (CTR) in *Saanen* dairy goats plus intramammary infusion in the right half of the mammary gland with *Strep. uberis* (Mastitis) while the left half of the mammary was left untouched (Control). Overall statistical significance (*p* ≤ 0.05) is indicated in the plot as follows: T = time effect; M = IMI effect; Z = effect of 2,4-thiazolidinedione injection. When a significant (*p* ≤ 0.05) interaction was present for T × M, * denotes the difference between groups at each time point.

**Figure 4 vetsci-06-00052-f004:**
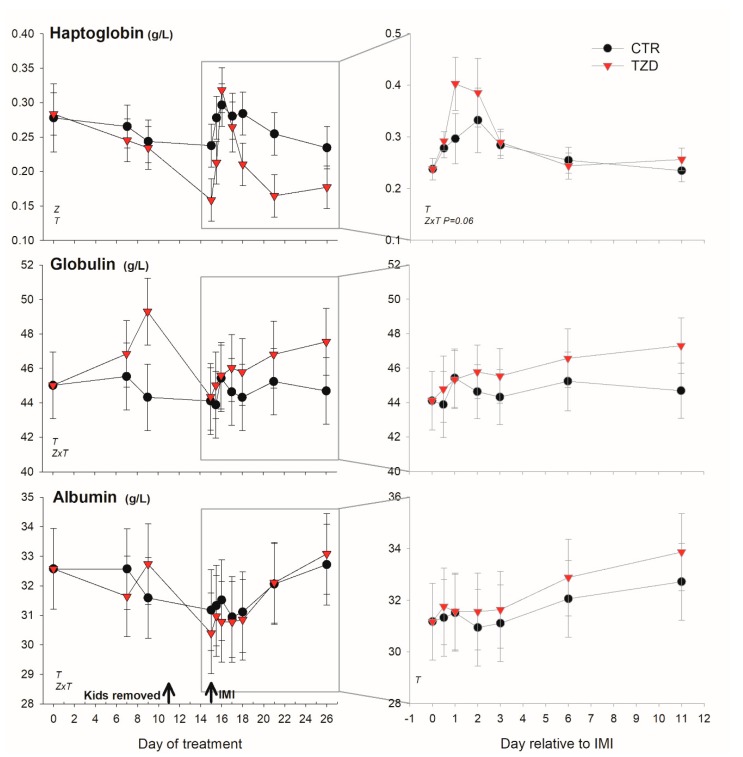
Effect on a positive (haptoglobin) and a negative (albumin) acute phase protein and globulin of daily injection of 2,4-thiazolidinedione (TZD) and saline (CTR) in *Saanen* dairy goats plus intramammary infusion in the right half of the mammary gland with *Strep. uberis* (IMI) while the left half of the mammary was left untouched (control). The left quadrants report the pattern of each parameter during the whole study. Arrows indicate time of kid removal and IMI. The quadrants on the right denote the pattern of the same parameters after IMI (with correction with baseline before IMI, see Materials and Methods). Overall statistical significance (*p* ≤ 0.05) is indicated in the plot as follows: T = time effect; M = IMI effect; Z = effect of 2,4-thiazolidinedione injection.

**Figure 5 vetsci-06-00052-f005:**
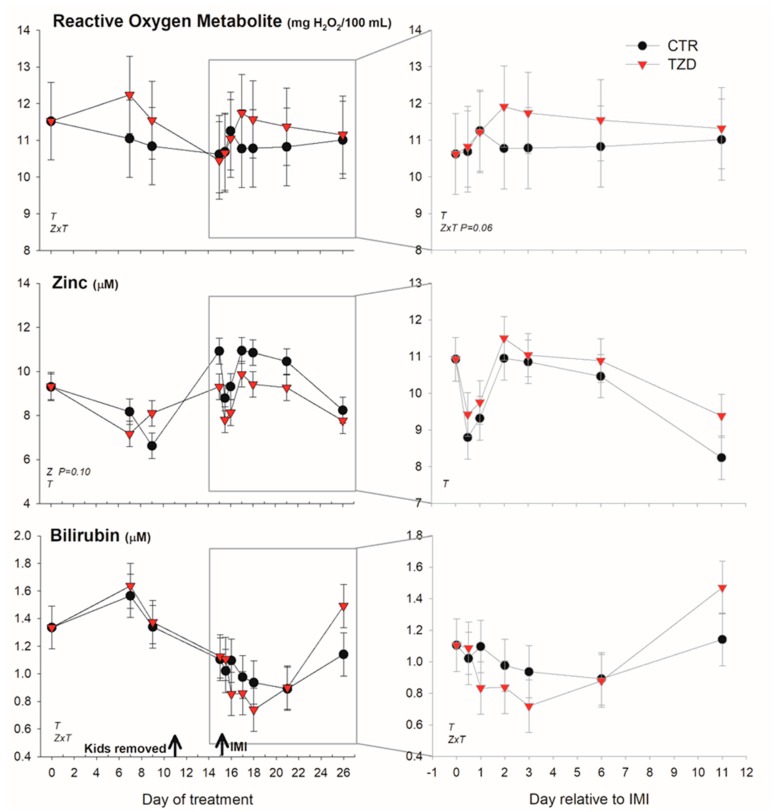
Effect of a daily injection of 2,4-thiazolidinedione (TZD) and saline (CTR) in *Saanen* dairy goats plus intramammary infusion in the right half of the mammary gland with *Strep. uberis* (IMI) while the left half of the mammary was left untouched (control) on reactive oxygen species, zinc, and bilirubin (as an index of liver function). The left quadrants report the pattern of each parameter during the whole study. Arrows indicate time of kid removal and IMI. The quadrants on the right denote the pattern of the same parameters after IMI (with correction with baseline before IMI, see Materials and Methods). Overall statistical significance (*p* ≤ 0.05) is indicated in the plot as follows: T = time effect; M = IMI effect; Z = effect of 2,4-thiazolidinedione injection.

**Figure 6 vetsci-06-00052-f006:**
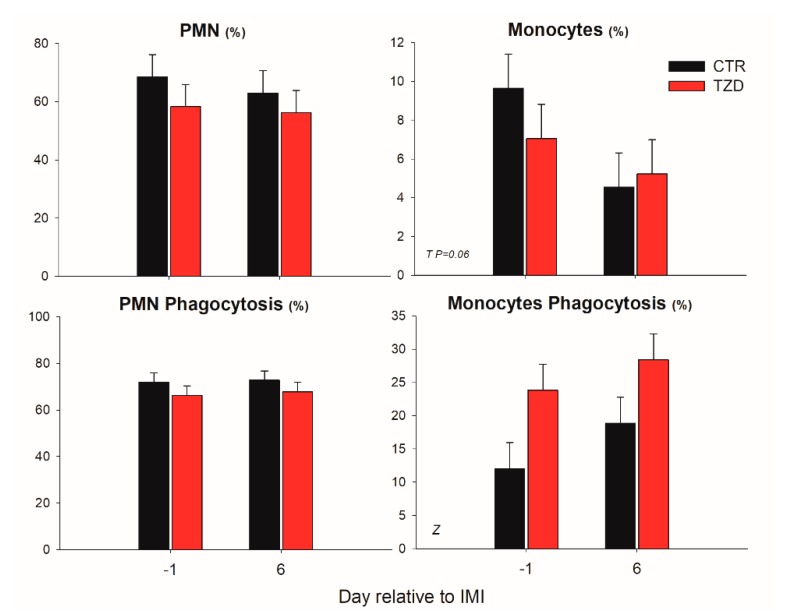
Effect on leukocytes differential and phagocytosis of 2,4-thiazolidinedione (TZD) daily injection and intramammary infusion (IMI) with *Strep. uberis* (IMI) in *Saanen* dairy goats. Reported are the percentages of granulocytes (PMN) and monocytes and their percent phagocytosis. Letters in the graph denote significant (*p* < 0.05) effects of time (T) and 2,4-thiazolidinedione treatment (Z).

**Figure 7 vetsci-06-00052-f007:**
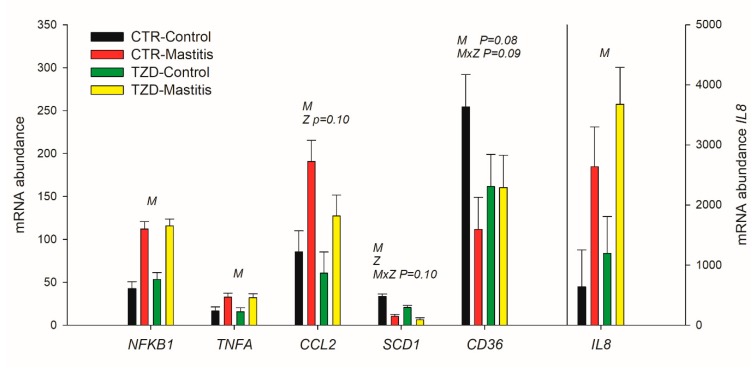
Effect on expression of inflammatory (*NFKB1*, *TNFA*, *CCL2*, and *IL8*) and milk fat synthesis (*SCD1*, *CD36*) related genes in mammary epithelial cells isolated from the milk of *Saanen* goats receiving either intrajugular daily injection of 2,4-thiazolidinedione (TZD) or saline (CTR) and intramammary infection in the right half of the mammary gland by infusing *Strep. uberis* (Mastitis) while the left half of the mammary gland was left untouched (Control). Cells were isolated 5 days post-intramammary infection. Letters in the plot denote overall statistical (*p* ≤ 0.05) effect for IMI (M), 2,4-thiazolidinedione treatment (Z), or interaction (M × Z). When a tendency was detected, the *p*-value is reported.

**Figure 8 vetsci-06-00052-f008:**
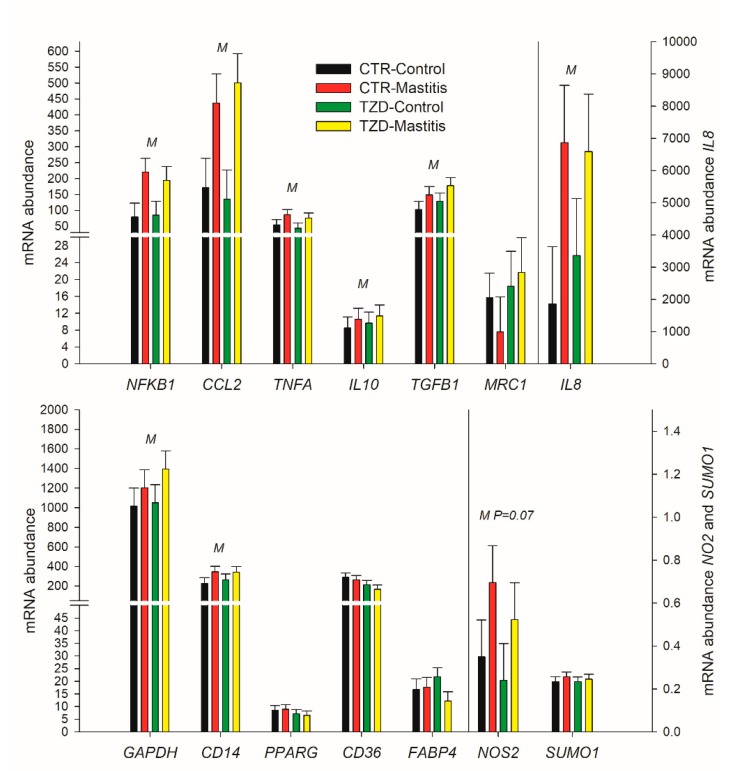
Effect on expression of genes related to classical (*NFKB1*, *TNFA*, *CCL2*, *NOS2*, and *IL8*) or alternative (*MRC1*) macrophages activation, macrophages deactivation (*IL10*, *TGFB1*), and metabolism in macrophage isolated from the milk of Saanen goats receiving either intrajugular daily injection of 2,4-thiazolidinedione (TZD) or saline (CTR) and intramammary infusion in the right half of the mammary gland of *Strep. uberis* (Mastitis) while the left half of the mammary was left untouched (Control). Cells were isolated 5 days post-intramammary infection (IMI). Letters in the plot denote overall statistical (*p* ≤ 0.05) effect for IMI (M), 2,4-thiazolidinedione treatment (Z), or interaction (M × Z). When a tendency was detected, the P-value is reported.

**Table 1 vetsci-06-00052-t001:** Blood metabolic parameters in goats treated with 2,4-thiazolidinedione (TZD) or saline (CTR) after infusion in the right gland of 7 × 10^8^ colonies of *Streptococcus uberis* (i.e., IMI).

Parameter	Group	Day from IMI							SEM ^1^	*p*-Value ^2^				
		0	0.5	1	2	3	6	11		Z	Time		Z × T	
Glucose (mM)	CTR	3.57	3.86	3.47	3.58	3.32	3.48	3.47	0.10	0.79	<0.01		0.58	
	TZD	3.57	3.72	3.56	3.52	3.36	3.50	3.33						
NEFA (mM)	CTR	0.07	0.05	0.08	0.06	0.07	0.06	0.11	0.03	0.61	<0.01		0.79	
	TZD	0.07	0.08	0.11	0.08	0.07	0.06	0.13						
TAG (mM)	CTR	0.19	0.16	0.17	0.18	0.19	0.19	0.13	0.02	0.97	<0.01		0.76	
	TZD	0.19	0.16	0.19	0.19	0.17	0.16	0.14						
BHBA (mM)	CTR	0.41	0.43	0.38	0.43	0.42	0.38	0.32	0.03	0.81	<0.01		0.27	
	TZD	0.41	0.40	0.43	0.41	0.35	0.36	0.34						
Urea (mM)	CTR	5.97	6.20	6.81	5.61	6.54	6.45	8.05	0.57	0.78	<0.01			0.35
	TZD	5.97	6.65	6.77	5.76	6.30	6.40	6.90						
Cholesterol (mM)	CTR	2.18	2.23	2.28	2.31	2.33	2.40	2.32	0.23	0.90	<0.01			0.88
	TZD	2.18	2.26	2.30	2.40	2.40	2.41	2.39						
Creatinine (μM)	CTR	71.9	70.0	72.6	70.2	70.4	72.2	75.8	2.5	0.69	<0.01			0.33
	TZD	71.9	71.2	73.4	72.9	71.8	73.6	77.3						

^1^ SEM= standard error of the LS-means; ^2^ Z= 2,4-thiazolidinedione effect, T = Time, Z × T = 2,4-thiazolidinedione × time effect.

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
