# Peer review of "2,4-Thiazolidinedione in Well-Fed Lactating Dairy Goats: II. Response to Intra-Mammary Infection"

_vetsci, 2019, doi:10.3390/vetsci6020052_

Round 1

Reviewer 1 Report

The manuscript is interesting but some improvements need.

1) lines 107-108: change this period as "Dry matter….are detailed in the previous manuscript."

2) line 114: "all the samples..pathogen" this is a result.

3) line 183: In my opinion a description of statistical analysis is mandatory. The authors cannot remand to a previous manuscript.

4) A statement on respect of animal welfare needs 

Author Response

We thanks the reviewer for the feedback and suggestions.

1) lines 107-108: change this period as "Dry matter….are detailed in the previous manuscript."

AU: Done as suggested

2) line 114: "all the samples..pathogen" this is a result.

AU: True it is a result, but was also a fundamental to have  animals that had not mastitis in order to enroll them into the study; thus, we would prefer to leave as it is.

3) line 183: In my opinion a description of statistical analysis is mandatory. The authors cannot remand to a previous manuscript.

AU: the statistical analysis is described. The reference to the prior publication is only regarding the data transformation.

4) A statement on respect of animal welfare needs 

AU: added the statement in section 2.1

Thanks!

Reviewer 2 Report

The authors revealed that PPARgamma agonist treatment in goats would function on immune system. Gene expressions involved in lipid and immune have been measured, with physiological  parameters. Given the fact that TZD has side effects in clinical research, did the authors detect the TZD concentration in milk and serum after treatment. It will be interesting if RNA-seq could be performed instead of qRT-PCR, as the author mentioned in the introduction, "nutrigenomics" will provide a genome-wide profile in this model in response to nutrition intervention like TZD or vitamin A. 

Author Response

Thanks for the comments. We agree with the  reviewer that a RNAseq analysis would be better, but it is pending financial support...for the detection of 2,4-TZD in milk and blood. We have actually done the analysis using HPLC. We have the data on the pharmacokinetics of 2,4-TZD (both after injection or oral) and the presence of 2,4-TZD in milk. The manuscript is in preparation.